# Variants of *STAR, AMH* and *ZFPM2/FOG2* May Contribute towards the Broad Phenotype Observed in 46,XY DSD Patients with Heterozygous Variants of *NR5A1*

**DOI:** 10.3390/ijms21228554

**Published:** 2020-11-13

**Authors:** Idoia Martínez de LaPiscina, Rana AA Mahmoud, Kay-Sara Sauter, Isabel Esteva, Milagros Alonso, Ines Costa, Jose Manuel Rial-Rodriguez, Amaia Rodríguez-Estévez, Amaia Vela, Luis Castano, Christa E. Flück

**Affiliations:** 1Biocruces Bizkaia Health Research Institute, Cruces University Hospital, UPV/EHU, CIBERER, CIBERDEM, ENDO-ERN. Plaza de Cruces 12, 48903 Barakaldo, Spain; idoia.martinezdelapiscinamartin@osakidetza.eus (I.M.d.L.); amaya.rodriguezestevez@osakidetza.eus (A.R.-E.); amaya.veladesojo@osakidetza.eus (A.V.); luisantonio.castanogonzalez@osakidetza.eus (L.C.); 2Department of Pediatrics, Endocrinology Section, Ain Shams University, 38 Abbasia, Nour Mosque, El-Mohamady, Al Waili, Cairo 11591, Egypt; dr.ranahakim@med.asu.edu.eg; 3Pediatric Endocrinology, Diabetology and Metabolism, Department of Pediatrics, Department of BioMedical Research, Inselspital, Bern University Hospital, University of Bern, Freiburgstrasse 15, 3010 Bern, Switzerland; kay.sauter@dbmr.unibe.ch; 4Endocrinology Section, Gender Identity Unit, Regional University Hospital of Malaga, Av. de Carlos Haya, s/n, 29010 Málaga, Spain; isabelestevadeantonio@gmail.com; 5Pediatric Endocrinology Department, Ramon y Cajal University Hospital, Ctra. de Colmenar Viejo km. 9, 100, 28034 Madrid, Spain; milagrosalonso2005@yahoo.com; 6Pediatric Department, Manises Hospital, Avda. Generalitat Valenciana 50, 46940 Manises, Spain; Icosta@hospitalmanises.es; 7Pediatric Endocrinology Department, Nuestra Señora de Candelaria University Hospital, Ctra general del Rosario 145, 38010 Santa Cruz de Tenerife, Spain; jmrial53@gmail.com; 8Pediatric Endocrinology Department, Cruces University Hospital, Plaza de Cruces 12, 48903 Barakaldo, Spain

**Keywords:** disorder/difference of sex development, DSD, steroidogenic factor 1, NR5A1/SF1, STAR, AMH, FOG2, oligogenic disorders, genotype–phenotype correlation

## Abstract

Variants of *NR5A1* are often found in individuals with 46,XY disorders of sex development (DSD) and manifest with a very broad spectrum of clinical characteristics and variable sex hormone levels. Such complex phenotypic expression can be due to the inheritance of additional genetic hits in DSD-associated genes that modify sex determination, differentiation and organ function in patients with heterozygous *NR5A1* variants. Here we describe the clinical, biochemical and genetic features of a series of seven patients harboring monoallelic variants in the *NR5A1* gene. We tested the transactivation activity of novel *NR5A1* variants. We additionally included six of these patients in a targeted diagnostic gene panel for DSD and identified a second genetic hit in known DSD-causing genes *STAR*, *AMH* and *ZFPM2/FOG2* in three individuals. Our study increases the number of *NR5A1* variants related to 46,XY DSD and supports the hypothesis that a digenic mode of inheritance may contribute towards the broad spectrum of phenotypes observed in individuals with a heterozygous *NR5A1* variation.

## 1. Introduction

Sex development depends on a tightly controlled network of transcription factors and signaling pathways working in concert to produce functional gonads and typical sex organs. Variations in this complex development lead to disorders/differences of sex development (DSD), which may manifest with a broad spectrum of clinical conditions [1]. The *NR5A1* gene, also known as the steroidogenic factor 1 (SF1) gene, is suggested as causative in 10–20% of 46,XY DSD [2]. A human variation in *NR5A1* was first detected in a 46,XY DSD individual with complete gonadal dysgenesis combined with adrenal insufficiency [3], and was mimicking the phenotype of the *Sf1* knockout (KO) mouse [4]. Later studies revealed that an adrenal phenotype is rarely found in patients with *NR5A1* variants [5], and that the gonadal and reproductive phenotype may be very broad including 46,XY and 46,XX cases [6]. Variable gonadotropins, sex hormones and anti-Müllerian hormone (AMH) levels are also found in patients with *NR5A1* variants, as well as inconsistent findings regarding gonadal histology [2].

*NR5A1* disease-causing variants are detected throughout the whole gene and they are mostly found in a heterozygous state. The pathogenic effect of *NR5A1* variants has been widely shown in clinical studies, however mechanistic proof through in vitro studies remains inconclusive. Transient in vitro transactivation assays of NR5A1 regulated genes did not show a functional impact of several variants, and a dominant negative effect for heterozygous variants was also not found [7]. In fact, haploinsufficiency seems to be insufficient to explain the complex phenotypic expression, as individuals with the same genetic variation may present with different clinical characteristics, even within the same family [8]. Given the highly varying phenotype and incomplete penetrance, a direct genotype–phenotype correlation is missing and needs further explanation [9]. Additional gene variants as disease modulators or environmental factors have been suggested [8]. Recently, Mazen et al. identified a second *MAP3K1* gene variant in a 46,XY DSD patient heterozygous for *NR5A1* [10]. Meanwhile several other patients have been reported carrying oligogenic *NR5A1* variants in combination with other variants in DSD-related genes [2,8,11,12,13].

The advantage of next generation sequencing (NGS) for elucidating the possibility of an oligogenic inheritance for DSD and other endocrine pathologies has been highlighted [14]. In the present work, we describe the clinical features of seven 46,XY DSD patients harboring a heterozygous *NR5A1* variation. We performed functional characterization of novel, possibly disease-causing variants. Additionally, individuals were investigated with a targeted gene panel including 48 DSD genes to search for a second hit that may contribute to their phenotype [15]. Finding likely disease-causing variants in known DSD genes in 50% of studied cases suggests that digenic inheritance may explain the variable phenotypic expression observed in these patients with heterozygous *NR5A1* variants.

## 2. Results

### 2.1. Clinical Features and Follow-Up

A summary of the clinical and biochemical features of the patients is given in Table 1. Three of the patients (Cases 1, 3 and 6) who were assigned male sex at birth presented with ambiguous genitalia (micropenis, hypospadias, bifid scrotum or cryptorchidism). Four patients were reared as females. Two of them (Cases 4 and 5) were referred at 14 years of age because of primary amenorrhea with rudimentary uterus and vaginal pouch. Additionally, Case 4 had clitoromegaly. The other two female patients (Cases 2 and 7) were referred at ages seven and one, respectively, for clitoromegaly and vaginal pouch. Case 2 also had bilateral inguinal masses.

All three male assigned individuals received surgical repair for their external genital ambiguity during early infancy. At follow-up, they were supplemented with testosterone, except for Case 1, who had spontaneous pubertal development. All cases maintained their assigned sex of rearing on follow-up. The three adult female patients who underwent gonadectomies (Cases 2, 4 and 5) and had hypoplastic/rudimentary uterus received estrogen replacement therapy.

### 2.2. Identification of NR5A1 Gene Variants and Other DSD-Related Gene Variants in Patients Presenting with 46,XY DSD

Pathogenic *NR5A1* gene variants according to ACMG (American College of Medical Genetics) standards [16] were identified in all seven patients either by analysis with a targeted gene panel or by candidate gene sequencing (Table 2). Case 1 carried a novel heterozygous c.88T>A missense variant in exon 2, leading to a p.Cys30Ser substitution in the loop of the first zinc finger domain of the protein (Figure 1). Both parents were studied and did not carry the variant, indicating that this change occurred de novo in the patient. Similarly, gene-by-gene analysis of Case 6 revealed another missense alteration in this crucial region of NR5A1; the His c.71A>T (p.His24Leu) change had not been reported before. Case 3 (male) presented with the c.250C>T; p.Arg84Cys *NR5A1* variant. Located in the A box of the DNA binding domain (DBD), this variant has been previously described in a case of 46,XY DSD [17]. Remarkably, his mother, maternal uncle and aunt suffer from DSD-related clinical features and were found to be carriers of the same heterozygous *NR5A1* variation. However, the same variant was also found in his healthy brother and grandfather. In Case 2, a frameshift insertion (c.614_615insC; p.Gln206ThrfsX20) in exon 4 was identified and predicted to produce an early protein truncation. Further *NR5A1* gene variations were detected in the ligand binding domain (LBD) of the protein (Figure 1). Case 5 and her mother presented the c.902G>A missense variant. Located in exon 5, this cysteine to tyrosine substitution in codon 301 (p.Cys301Tyr) is novel. Finally, in Case 4, a four-nucleotide deletion (c.910_913delGAGC) was found and predicted to produce a shorter protein consisting of 330 amino acids (p.Glu304CysfsX26) instead of 461 as seen with wild-type (WT) NR5A1. The mother of Case 4 is a carrier of the variation. Sequencing of Case 7 revealed a heterozygous c.1183_1185delGAG non-frameshift variation in exon 7 that leads to the deletion of amino acid 395 (p.Glu395del) in the LBD of the protein. This variant was also found in the mother of the patient.

Except for Case 4, DNA samples of patients, in which only a single gene approach had been performed initially, were further analyzed by the targeted DSD gene panel including 48 genes [15]. Additional heterozygous variants in known DSD genes were found in three out of six (50%) 46,XY DSD *NR5A1* carriers. For Case 1, a novel missense VUS (variant of unknown significance) variant (c.361C>T; p.Arg121Trp) in the *STAR* gene was identified in the patient and his father. A rare variant in *AMH*, c.428C>T; p.Thr143Ile, was detected in Case 2 and was classified as VUS. Finally, for Case 7 and her father, a previously reported *ZFPM2/FOG2* (c.1632G>A; p.Met544Ile) pathogenic missense alteration was identified [18]. No additional variants in other genes associated with testicular development and DSD were identified by our panel in Cases 3, 5 and 6. We were not able to further analyze Case 4 because of a lack of sample availability.

### 2.3. Transcription Activity and Protein Expression Testing of Novel NR5A1 Variants

To study the impact of the three novel missense variants of *NR5A1* on transactivation activity of regulated genes, HEK293 cells were co-transfected with WT or mutant NR5A1 expression vectors and three different promoter reporter constructs essential for steroid and sex hormone biosynthesis. All three novel NR5A1 variants had significantly reduced activity on the CYP17A1 reporter compared to WT (Figure 2A). These results were confirmed for the His24Leu and Cys30Ser variants when using the reporters for CYP11A1 and HSD17B3 (Figure 2B,C). In contrast, variant Cys301Tyr did not change the reporter activities of CYP11A1 and HSD17B3 (Figure 2B,C).

Expression of NR5A1 variants was assessed by Western blot in our cell model. As shown in Figure 2D, SF1 protein expression was similar for all studied NR5A1 variants.

## 3. Discussion

Patients harboring *NR5A1* variants manifest with extremely broad phenotypes, ranging from normal sex development to complete sex reversal. A lack of genotype–phenotype correlation has been widely questioned, and the hypothesis of additional genetic variations contributing to the complex and variable phenotype has been formulated [2,8,10,13]. In this work, we report clinical and genetic data of seven 46,XY DSD patients with heterozygous *NR5A1* variants with normal adrenal function. Using targeted gene panel analysis for sex development-related genes, we found a second likely disease causing/pathogenic variant in known DSD genes (*STAR*, *AMH*, *ZFPM2*) in three of six studied 46,XY DSD individuals, supporting the hypothesis of oligogenic disease.

*NR5A1* variants are often found in 46,XY DSD individuals. Their external genital phenotype may vary from typical male with infertility to hypospadias or ambiguous genitalia, to typical female external phenotype with primary amenorrhea [2]. Likewise, our patients presented with a wide clinical spectrum (Table 1). Three patients manifested at birth as undervirilized males with ambiguous genitalia and four presented with typical female external genitalia. They mostly had adequate testosterone production and good or even high response to hCG stimulation tests, even enough to induce pubertal development in some of them. According to the literature, variable hormonal findings are found in persons with *NR5A1* variants and do not correlate with the phenotype of the external genitalia [2]. Normal testosterone concentrations, at least in early childhood, have been found in *NR5A1* variant patients with cryptorchidism, hypospadias and micropenis [7,20], as well as in 46,XY subjects presenting with a female phenotype [21,22]. In contrast, testosterone synthesis might not be sufficient to start or proceed through puberty. This may be explained by a progressive replacement of the differentiated normal testicular tissue with connective tissue over time, followed by the loss of seminiferous tubules, finally leading to the absence of gonadal tissue in adulthood [7]. As expected, the two male patients (Cases 3 and 6) with normal hCG stimulation tests in early childhood needed testosterone supplementation for pubertal development. Interestingly, Case 1 achieved spontaneous pubertal development, and the female patients presented with signs of (progressive) virilization at different ages. Virilization [23] or spontaneous pubertal development [2] have been described in 46,XY DSD individuals with a *NR5A1* variant, even with elevated gonadotropin levels [7], indicating that the Sertoli cell damage might be more severe than the damage to the Leydig cells [21]. Our 46,XY DSD cases had generally low AMH levels, which might explain why the three female assigned persons had a rudimentary developed uterus, and fallopian tubes were only found in one. However, similar to previous cases [20,21,24], no Müllerian structures were found in the other four studied cases (one female, three males) despite only low AMH. This might be explained by the prominent action of AMH in utero, and with progressive malfunction of the AMH-producing Sertoli cells with age, evidenced by the gradually increasing FSH with low LH/FSH ratio [21,24]. None of our patients had symptoms of adrenal insufficiency so far. Although adrenal insufficiency is a rare finding in individuals with *NR5A1* variants [6], adrenal function should be followed regularly as life-threatening adrenal insufficiency may develop later in life [2].

Huge phenotypical variability has also been observed within families carrying *NR5A1* variants [7,25]. We confirmed *NR5A1* variant transmission from one of the parents in four of six cases. In two cases, no carriers were found (Table 2). Family history of one of our index patients revealed premature menopause, menstrual disorders and isolated hypospadias in three relatives harboring the same p.Arg84Cys *NR5A1* variation. In 46,XX individuals, *NR5A1* variants are associated with primary ovarian insufficiency [7,26], and most of these women have relatives with a 46,XY DSD [27]. Similarly, *NR5A1* sequence variants have been described in males with isolated hypospadias [28]. The majority of reported *NR5A1* variants are maternally inherited or occurred de novo [29]. Four of our studied cases (Cases 3, 4, 5 and 7) showed maternal inheritance. In the two cases where variant inheritance was not proven, only one of the parents was available for testing for one case (Case 2); thus we cannot determine whether the p.Gln206ThrfsX20 variant occurred de novo, although the exactly same variation has been described [7]. Healthy carriers with heterozygous variants in *NR5A1* have been reported [7].

Previously reported *NR5A1* variants were mostly non-synonymous and were mainly located in the DBD of the protein [6]. We detected two novel variants in the first zinc finger of the DBD (His24Leu, Cys30Ser) that manifested with undervirilized ambiguous male genitalia at birth and with elevated gonadotropins and small testes later. In addition, one case (Case 6) needed androgen replacement early in life and was proven to have Leydig cell hypoplasia. Camats et al. [7] described other amino acid changes in these codons (His24Tyr and Cys30Trp) in 46,XY DSD females with an inguinal hernia and abnormal testicular tissue. The highly conserved residues in these domains are essential for DNA and zinc binding (Cys30), and genetic alterations probably interfere with binding and inhibit transcriptional activation, resulting in variable loss of NR5A1 function. A third variant (Arg84Cys), identified in one of our cases, was located in the A-box of the DBD crucial for DNA anchoring. This pathogenic variant has been described in 46,XY males with similar phenotypes [17,30], but also in a 46,XY female with adrenal insufficiency [31], which confirms the clinical divergence of a single genotype with NR5A1 DSD. Case 2 carried the reported Gln206ThrfsX20 variant in the hinge region of the protein, predicted to produce an early truncation and ineffective NR5A1 protein. The prior reported patient with this frameshift variation had similar clinical features, except for the hypoplastic uterus [7]. The hinge region of NR5A1, besides connecting the LBD and the DBD, is important for protein configuration and is a target for post translational modifications, such as phosphorylation, which has been demonstrated to promote NR5A1 stability and transcriptional activity [32]. The LBD is responsible for modulating NR5A1 activity through bond formation between its AF2 domain and phospholipid ligands [33]. We identified three novel *NR5A1* variants in the LBD in female phenotypic 46,XY DSD cases. Two variants are predicted to produce truncated proteins (Glu304CysfsX26 and Glu395del) and one a missense Cys301Tyr protein. While our 46,XY cases with NR5A1 variations in the LBD presented with a predominant female phenotype, others have reported cases with variants in same and similar location with a rather male typical phenotype, e.g., hypospadias and anorchia [34,35].

When tested for promoter transactivation activity in HEK293 cells, we found that the two NR5A1 variants located in the DBD showed consistently impaired activity, while Cys301Tyr, located in the LBD, demonstrated variable results with different promoters, and therefore its pathogenicity may be questioned. Similarly inconsistent results have been reported for other variants [2,7,21].

Possible digenic or oligogenic inheritance of genetic disorders of sex development with a broad phenotype has been discussed by us [8,14,36] and others [2,10,11,12,13,37,38]. We therefore searched for additional gene variants in the studied cases and found a second genetic variant in a gene previously reported to be involved in male sex development [6] in three individuals (Table 2). In two cases (Cases 2 and 3), NGS analysis revealed missense monoallelic variants in the *STAR* (OMIM 600617) and *AMH* (OMIM 600957) genes, respectively. Although the identified Arg121Trp *STAR* variant was defined as pathogenic by most prediction tools, no *STAR* variant has been described with an isolated DSD phenotype so far, and with a disease phenotype at all in heterozygote carriers. In contrast, mild autosomal recessive, non-classic *STAR* genetic alterations may cause isolated adrenal insufficiency [39]. Thus, the Arg121Trp *STAR* variation must be categorized to be of unknown significance (VUS). Interestingly, the same heterozygous *STAR* variant was found in the healthy father, who is not carrying the *NR5A1* variant (shared by the mother), suggesting that only the combination of both gene variants may lead to the observed phenotype. Certainly, it is advisable to follow this patient closely for the development of adrenal insufficiency. Concerning the rare variant in the *AMH* gene identified in Case 2, it has been previously described in a female with polycystic ovary syndrome, as well as in a control population [19]. Finally, the female Case 7 and her father carried a heterozygous *FOG2*/*ZFPM2* missense variation (OMIM 603693). Similar to the previously reported cases by Bashamboo et al. [18], our patient presented with 46,XY gonadal dysgenesis, no heart anomalies, and had genetically positive relatives who were asymptomatic. The potential role of heterozygote variations in the *ZFPM2* gene might therefore only be relevant in combination with a variation in another DSD-related gene.

It is well known that both *STAR* and *AMH* genes are regulated by NR5A1 [6,40]. The nuclear receptor NR5A1/SF1 is expressed in the bipotential gonad, and modulates genes like *SRY* and *SOX9* to regulate and maintain male determination [41]. In Sertoli cells, NR5A1/SF1 enables the expression of *AMH* and its receptor *AMHR2*, which are essential for the regression of the paramesonephric ducts [42]. NR5A1/SF1 also regulates the expression of genes essential for steroidogenesis, such as *STAR* and *CYP17A1* in Leydig cells, which are required for testosterone biosynthesis and virilization of the fetus [33]. The zinc finger protein ZFPM2 (or FOG2, Friend of Gata) interacts with GATA4 to modulate its transcriptional activity. Formation of heterodimers together with NR5A1/SF1 is thought to be necessary for proper cell differentiation in the testis [43], and has been demonstrated to regulate the expression of Sry, Sox9 and Amh in mice [44,45,46]. Although initially described in patients with cardiac anomalies only [47], recent studies have illustrated that haploinsufficiency and variants of the *ZFPM2/FOG2* gene may cause 46,XY gonadal dysgenesis [43]. Furthermore, the ZFPM2/FOG2 variant Met544Ile showed impaired interaction with GATA4 [18]. More recently, all previously described variants of *ZFPM2/FOG2* were revaluated and classified as benign due to their allelic frequencies and transactivation activity similar to WT [48]. Therefore, we and others [8] suggest that heterozygous *ZFPM2/FOG2* variants may not be pathogenic by themselves, but contribute to a DSD phenotype when occurring in combination with heterozygous *NR5A1* variants (or in combination with variants of other DSD-related genes). With this hypothesis in mind, the normal phenotype of the fathers of Cases 1 and 7 compared to their offspring can be explained by the presence of variants in more than one gene in the patients, producing a cumulative pathogenic effect.

DSD is caused by a large group of genetic conditions and the discovery of underlying genes is constantly expanding through the use of NGS technologies. In this study, we have characterized seven patients harboring novel and known heterozygous *NR5A1* variants and thereby increased the number of known *NR5A1* variants. Using a DSD targeted gene panel, we found additional variants in *STAR, AMH* and *ZFPM2* in three of the six studied individuals with variable DSD phenotypes. This clearly supports the hypothesis that the broad clinical spectrum seen with *NR5A1* variants is due to di-/oligogenic causation. Overall, the genetics underlying DSD might be more complex than initially thought. Variable phenotypes and lack of a genotype–phenotype correlation hint at a complex network of genes and modifiers working together. The use of NGS offers the opportunity to explore this complex network. However, to understand the role of each participating gene and the mechanistic connections remains challenging. Therefore, in future work we plan to study the specific role of identified genes involved in this network using induced steroidogenic cells reprogrammed from patient fibroblasts carrying the genetic signature of the individual patient. Basic protocols for this novel approach to perform mechanistic studies and move towards cell based therapies have been published recently [49,50,51].

## 4. Materials and Methods

### 4.1. Patients and Samples

The seven patients with 46,XY DSD included in this study are part of a larger cohort recruited at the Biocruces Bizkaia Health Research Institute (Barakaldo, Spain). Clinical data were provided by the responsible clinicians and a written informed consent was obtained from all participants or their parents. The study was approved by Swiss ethics (BASEC 2016-01210, date: 5.11.2017) and informed by the local ethical committees, namely the ethical boards of the Regional University Hospital Malaga, Spain, Ramon y Cajal University Hospital, Madrid, Spain, Manises Hospital, Manises, Spain, Nuestra Señora de Candelaria University Hospital, and Santa Cruz de Tenerife, Spain, as well as the ethics committee for clinical research of Euskadi (CEIC-E), Spain. Additional studies on this cohort have been reported elsewhere [52].

### 4.2. Case Reports

A summary of the clinical and biochemical characteristics of the seven studied patients is given in Table 1.

Case 1: This newborn was noted at birth to have curved micropenis, chordee with scrotal hypospadias and bilateral cryptorchidism. Surgical correction was performed. Karyotype was 46,XY. At 10 years of age, stimulation test with human chorionic gonadotropin (hCG) resulted in a normal rise of testosterone (187.4 ng/dL). At 12 years of age, his penis was 7.6 cm in size, and testes nearly 5 mL. MRI showed testes of 3.2 mL and 3.3 mL. Cysts of 6.4 mm and 2.7 mm were observed in the right and left epididymis, respectively. Asperger syndrome was suspected at the age of 13 years old, but the genetic study of the *FMR1* gene was normal. At 14 years of age, biochemical analysis revealed elevated serum levels of gonadotropins (LH 11.4 U/L, FSH 35.9 U/L) and a testosterone level of 412.5 ng/dL with low levels for AMH (<0.1 ng/mL). At a recent follow-up at 15 years of age, penile size is still in the normal limit with high levels of LH and FSH with a testosterone level of 518 ng/dL. Family history is remarkable for unilateral renal agenesis in the mother as well as in a brother, who also has vas deferens and left epididymis agenesis.

Case 2: This female was referred at 10 years of age because of clitoromegaly, bilateral inguinal masses and accelerated growth. She presented with early pubarche and facial hair growth. Karyotype was 46,XY. Pelvic ultrasound showed a vaginal pouch and absent uterus. Palpable gonads were located at the inguinal canal (2 and 3 mL). Biochemical analysis was remarkable with elevated gonadotropins and a testosterone of 250 ng/dL. Gonadectomy and histology revealed two testes of 2.4 and 2.5 cm with germ cell hypoplasia. Oral estrogen therapy was initiated and replaced by transdermal supplementation after 9 years. This resulted in good breast development, although regression was observed when the dose was decreased. During the following years, reconstructive surgery was performed, achieving a permeable vagina of 2–3 cm and a normal clitoris. Abdominal ultrasound showed a normal bladder and a small structure (26 mm) above the urethra, hinting at a possible prepuberal hypoplastic uterus. No ovaries were visualized. At 27 years of age, the patient was well developed. Hormonal values were in the normal, age- and sex-appropriate range under supplementation therapy (LH, E2). FSH remained elevated after gonadectomy.

Case 3: This patient presented with micropenis, scrotal hypospadias, undescended testes, and bifid scrotum at birth. Karyotype was 46,XY. Abdominal ultrasound revealed no anomalies. Biochemically, LH and FSH were normal (2.5 and 3.7 U/L, respectively) with a low testosterone of 67.5 ng/dL in the first month of life. At 3 months of age, four doses of testosterone were prescribed (50 mg/dose) with adequate penile response. When he was 4 years old, he had hypospadias repair surgery. At 5 years of age, physical examination showed a penis of 2.5 cm in size, right testis of 2 mL and a hydrocele of the left scrotal compartment. At 6 years of age, replacement therapy with testosterone was started. Penile length increased to 3 cm and pubic hair was noted. Interestingly, his mother presented with premature menopause, the maternal uncle presented with scrotal hypospadias at birth, and his aunt presented with menstrual disorders.

Case 4: This female was seen at 14 years of age because of virilization during puberty and primary amenorrhea. Pubarche and axilarche had started at 8 years of age. The patient presented with typical female genitalia at birth. Physical exam revealed a female clinical phenotype with excessive facial and body hair and lack of thelarche. Clitoris with perineal-located urethra were noted. There were no inguinal masses. Karyotype was 46,XY. Abdominal and pelvic MRI showed a rudimentary uterus, and a computerized axial tomography (CAT) scan showed a small uterus and atrophic gonads in the left inguinal canal. Laboratory analysis showed elevated LH and FSH levels (37.8 U/L and 113 U/L respectively) with high levels of testosterone for a female (193 ng/dL). Antiandrogens were prescribed (50 mg/day). At 15 years of age, gonadectomy was performed. Histology revealed fibro-adipose and skeletal muscle tissue together with a testis of Sertoli-cell only phenotype and well identified epididymis. After surgery, estrogen patches were added to the pharmacological treatment.

Case 5: This patient presented with primary amenorrhea and obesity at 14 years of age. She had a stenotic and enlarged vagina, which was possibly a vaginal pouch. Karyotype was 46,XY. Abdominal and pelvic ultrasound and MRI demonstrated a rudimentary uterus and vagina, but neither testes nor ovaries were identified. Biochemical analysis at this time showed increased levels of gonadotropins with a testosterone level of 36.6 ng/dL only. Stimulation test with hCG resulted in an insufficient rise of testosterone (47.70 and 46.0 ng/dL for the third and fifth dose of hCG, respectively). Laparoscopy at 15 years of age revealed a rudimentary uterus and streak gonads. Gonadectomy and biopsy showed gonadal tissue compatible with non-functioning testicular parenchyma and normal fallopian tubes. Family history was unremarkable.

Case 6: This patient was noted at birth to have curved penis buried in mons pubis, perineal hypospadias, bifid scrotum and palpable gonads. His karyotype was 46,XY. At 2 years of age, biopsy showed testicular tissue without alterations, and laparoscopy did not find Müllerian remnants. During the following years, hypospadias was corrected. At 10 years of age, penile length was 2.6 cm, right testis was located in scrotum and the left was inguinal. Stimulation test with hCG resulted in a normal rise of testosterone. He started treatment for micropenis with topical application of testosterone, which increased phallic size to 3.5 cm and initiated pubarche. At 14 years of age, the boy had a penile length of 5.5 cm and testicular volume of 2–3 mL bilaterally. Biochemically, gonadotropins were elevated (LH 15 U/L; FSH 55 U/L) and testosterone low (1.8 ng/dL). ACTH, cortisol and 17-hydroxyprogesterone were within the normal range. Bone age was similar to chronological age. Intramuscular testosterone treatment was prescribed again (125 mg/3 weeks) for 6 months. At 15 years of age, penile length was 7 cm and testes were still only 2 mL in volume. Testosterone treatment was stopped for 2 months and LHRH stimulation test revealed a normal response of gonadotropins and testosterone (Table 1). In the following years, orchidopexy was performed, and a second biopsy revealed the absence of germ cells, and Leydig cels hypoplasia. Thus, testosterone supplementation treatment was restored, but stopped again shortly after by the patient because of widespread edema. Currently, the patient has a penile length of 3–4 cm and bilateral testicular volume of 2 mL. LH and FSH are elevated and testosterone is low (500 ng/dL). The patient also suffers from obesity (BMI: 42 kg/m^2^).

Case 7: This 1-year-old female was referred to paediatric surgery for clitoromegaly and fused labia minora. At that time, ultrasound was not conclusive and abdominal and pelvic MRI showed a small vagina (3.2 cm) without fallopian tubes. Karyotype was 46,XY. Basal hormonal testing indicated slightly reduced FSH (1.23 U/L) while LH was elevated (0.56 U/L). Testosterone levels were in the normal female range, and AMH was increased (40.3 ng/mL). After stimulation with hCG, testosterone increased highly (8.0 ng/dL), while DHEA-S levels remained in the normal range for age and female sex. During genitoplasty, the absence of a uterus and ovaries was confirmed. When the patient was 3 years old, a structure of 3 cm was seen by ultrasound, most likely corresponding to a vaginal pouch. Recent laboratory results at 7 years of age showed normal values for age and sex (Table 1).

### 4.3. Genetic Testing

Extraction of DNA from peripheral blood leukocytes was performed using either the MagPurix Blood DNA Extraction Kit 200 (Zinexts Life Science Corp., New Taipei City, Taiwan) or the QIAamp DNA Blood Minikit (Qiagen, Venlo, Netherlands).

### 4.4. Design of the Targeted Gene Panel and Screening via Next-Generation Sequencing

A targeted gene panel (TGP) was designed to include 48 genes associated with DSDs in online databases, including PubMed (https://www.ncbi.nlm.nih.gov/pubmed/), Human Gene Mutation Database (HGMD, https://portal.biobase-international.com) and Online Mendelian Inheritance in Man (OMIM, https://www.ncbi.nlm.nih.gov/omim) [15]. Libraries were prepared according to the manufacturer’s instructions and samples were sequenced using the Ion PGM platform (Thermo Fisher Scientific, Waltham, MA, USA). Base calling, read filtering, alignment to the reference human genome GRCh37/hg19, and variant calling were done using Ion Torrent Suites (Thermo Fisher Scientific, Waltham, MA, USA). Further QC analysis, coverage analysis and variant filtering were completed with the Ion Reporter Software (Thermo Fisher Scientific, Waltham, MA, USA). Coverage depth and read quality were evaluated with the Integrative Genomics Viewer (IGV, broadinstitute.org, Cambridge, MA, USA). Variants were filtered to include only those with a Phred-like score ≥ 30, *p*-value < 0.001, and minor allele frequency (MAF) < 1%. Variants found in the TGP were confirmed via amplification of the exons by PCR and sequencing. When available, DNA samples from the patients’ relatives were screened likewise.

### 4.5. Sequencing of the Coding Sequence of NR5A1

The coding exons of *NR5A1* were amplified by PCR and analyzed by bidirectional direct sequencing using the BigDye Terminator v3.1 Cycle Sequencing Kit (Thermo Fisher Scientific, Waltham, MA, USA) in an ABI 3130xl DNA Genetic Analyzer (Thermo Fisher Scientific, primer sequences and conditions are available from the authors, on request). All variants were annotated according to the GenBank reference sequences NM_004959 and NP_004950.

### 4.6. In Silico Analyses and Variant Classification

We used the following online tools to predict the impact of the non-synonymous alterations on protein function: PROVEAN (Protein Variation Effect Analyzer) (http://provean.jcvi.org/index.php), SIFT (Sorting Intolerant From Tolerant) (http://sift.jcvi.org/), Polyphen 2 (http://genetics.bwh.harvard.edu/pph2/), Mutation Taster (http://www.mutationtaster.org/), SNPs and Go (http://snps.biofold.org/snps-and-go/snps-and-go.html), MutPred (http://mutpred.mutdb.org/), Panther (http://www.pantherdb.org/) and Varsome (https://varsome.com/). Variants were classified following the standards and guidelines from the American College of Medical Genetics and Genomics (ACMG) [16]. Previously reported clinical associations were searched in the databases ClinVar (https://www.ncbi.nlm.nih.gov/clinvar) and HGMD.

### 4.7. In Vitro Transactivation Assays and Protein Expression Analysis

Non-steroidogenic human embryonic kidney HEK293 cells (ATCC CRL-1573) were cultured and used for functional assays with promoter luciferase reporter vectors –227CYP17A1_Δluc, –152CYP11A1_pGL3 and –301HSD3B2_pGL3 as previously described [7,53]. NR5A1 expression vectors containing the novel c.88A, c.902A or c.71T variants were generated by site-directed mutagenesis using specific primers and the QuickChange protocol by Stratagene (Agilent technologies Inc., Santa Clara, CA, USA). Results are shown as the mean ± SEM of five independent experiments, performed in duplicate. Data were statistically analyzed using the Student’s t test. A *p*-value  ≤  0.05 was considered significant.

Expression level of WT and mutant SF1 proteins in HEK293 cells after transfection was assessed by Western blot analysis of cell lysates with an antibody against HA-tag (Merck KGaA, Darmstadt, Germany). Expression of β-actin protein was used as control.

## Figures and Tables

**Figure 1 ijms-21-08554-f001:**
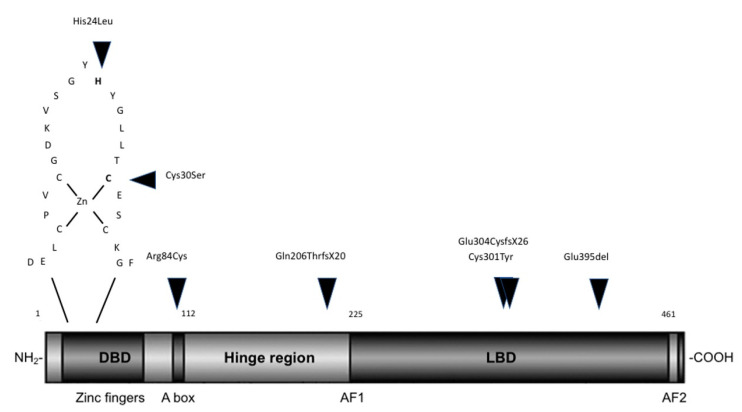
Scheme of the structure of the NR5A1 protein and localization of studied variations. Critical functional domains comprise a DNA-binding domain (DBD) at the amino terminal (from amino acids 13 to 112) containing two zinc finger domains (ZNI and ZNII) and an A box, the flexible hinge region (amino acids 112–225) and a ligand-binding domain (LBD) (amino acids 225–458) with activation function 1 (AF1) and 2 (AF2). A close-up loop of the first zinc finger domain in DBD is shown where the His24Leu and Cys30Ser alterations are located. Locations of the NR5A1 variations described in this work are indicated by arrow heads.

**Figure 2 ijms-21-08554-f002:**
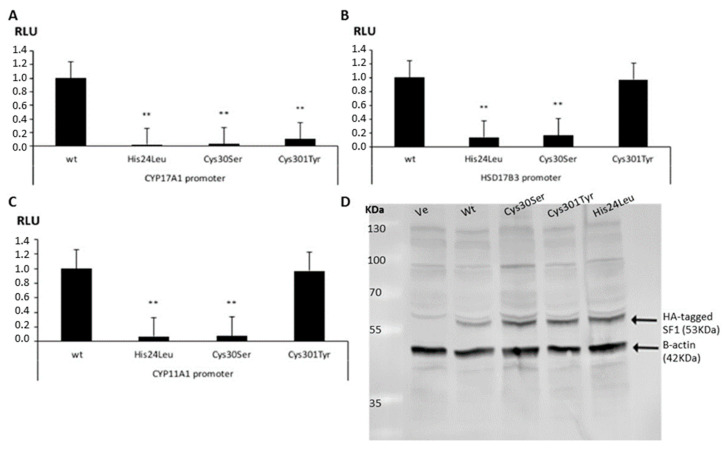
Functional studies of the three novel NR5A1 variants. The ability of wild-type (WT) and mutant NR5A1 to activate promoter luciferase reporter constructs was tested in HEK293 cells (**A**–**C**). Cells were transiently transfected with NR5A1 expression vectors and promoter reporter constructs: 227CYP17A1_Δluc (**A**), -301HSD17B3_pGL3 (**B**) and -152CYP11A1_pGL3 (**C**) [7]. Results are shown as the mean ± standard error of the mean (SEM) of five independent experiments, all performed in duplicate. (**D**) Western blot showing expression of WT and mutant NR5A1/SF1 proteins. The HA antibody recognized hemagglutinin-tagged NR5A1/SF1 (band at 53 kDa). β-actin was used as a control (band at 42 kDa). ** *p*-value  ≤  0.01. HA, hemagglutinin; RLU, relative light units; Ve, empty vector; WT, wild type.

**Table 1 ijms-21-08554-t001:** Clinical and biochemical characteristics of the studied patients.

Case (Gender)	Phenotype	Hormonal Findings									
	Genital Anatomy, Histology and Presence of Müllerian Ducts	Age	17OHP	DHEA-S	Androst	T	DHT	FSH	LH	AMH	Others
1 (m)	At birth: curved micropenis, hypospadias, cryptorchidism. Surgery. At 12 y: penis 7.6 cm. MRI: testis 5 mL, small cysts. At 15 y: MRI: right testis (3.2 mL) and left testis (3.3 mL), small cysts. Absence of Müllerian ducts.	10 y			1.7/1.9 †	86.3/187.4 †	0.19/0.15 †	11.9	0.9		
		11 y	1.3		2.5	290.5	0.5	30.4	7.1		
		14 y		1900		**412.5**		**35.9**	**11.4**	**<0.1**	
		15 y		2010	4.1	518	0.5	**38.9**	**14.6**	**0.4**	
2 (f)	At 10 y: bilateral inguinal hernia and gonads, clitoromegaly. US: vaginal pouch, no uterus. Histology: hypoplastic testis. Gonadectomy. At 20 y: good breast development after estrogen treatment. At 25 y: US: prepuberal hypoplastic uterus. No ovaries. Müllerian ducts.	10 y	1.9		**1.0**	**250**		**24**	**95**		E2: 16
		27 y	0.5		2.5			**21.4**	8.4		E2: 26.4
3 (m)	At birth: micropenis, hypospadias, undescended testes. 5 y: normal penis, right testis 2 mL and hydrocele left testis. At 6 y: smaller penis, scrotal testes. Absence of Müllerian ducts.	2 d		202		**23.6**	1.5	<0.5	<0.5		F: 10.4
		1 m		400		**67.5**		3.7	2.5		F: 6.1
4 (f)	At 14 y: primary amenorrhea, virilization. clitoromegaly, scrotal rests. US: rudimentary uterus, atrophic testis in left inguinal canal. At 15 y: gonadectomy. Histology: testis with Sertoli cells and epididymis. Müllerian ducts: rudimentary uterus.	14 y	**0.9**	1.61	1.7	**193**	0.5	**112.5**	**37.8**	**0.4**	E2: 16.7
5 (f)	At 14 y: primary amenorrhea, stenotic and enlarged vagina. At 15 y: laparoscopy: rudimentary uterus, streak gonads. Gonadectomy. Histology: testicular parenchyma, normal fallopian tubes. Müllerian ducts: vagina, rudimentary uterus.	14 y	0.5	1960	2.0	**36.6**/46 †		**44.3**	**12.2**	<0.01	F: 18/27 †; E2: 31.0
6 (m)	At birth: micropenis, hypospadias, palpable gonads. At 2 y: biopsy. Histology: testicular tissue. 17 y: absence of germ cells and Leydig cells hypoplasia. At 10 y: penis 2.6 cm. At 14 y: penis 5.5 cm, scrotal testes. At 17 y: biopsy. At 38 y: penis 3–4 cm, testes 2 cc. Absence of Müllerian ducts.	10 y				20/130 †					
		14 y	N			180/350 †		**55**	**15**		ACTH: N; F: N
		15 y				**140**/500 †		**39**/72 †	**11**/137 †		
		38 y	0.3	1293	1.4	**500**		**25**	**15.6**	<0.1	ACTH: 36; F: 14.3; E2: 20
7 (f)	At 1 y: hypertrophic clitoris, fused labia minora. MRI: vagina (3.2 cm) without annexes. Genitoplasty. At 3 y: US: absence of ovaries and uterus. Absence of Müllerian ducts.	1 y	0.4	<17	0.6	8		1.2	0.5	40.3	F: 12.1; E2: <10
		7 y				14		3.8	0.05		E2: <11

17OHP, 17-hydroxyprogesterone (ng/mL); ACTH, adrenocorticotropic hormone (pg/mL); AMH, anti-Müllerian hormone (ng/mL); Androst, androstenedione (ng/mL); F, cortisol (ug/dL); DHEA-s, dehydroepiandrosterone sulphate (ng/mL); DHT, dihydrotestosterone (ng/mL); E2, estradiol (pg/mL); f, female; FSH, follicle-stimulating hormone (U/L); LH, luteinizing hormone (U/L); m, male; MRI, magnetic resonance imaging; N, normal; T, testosterone (ng/dL); US, ultrasound; y, years. † After human chorionic gonadotropin (hCG) stimulation test. All patients presented with a 46,XY karyotype. Laboratory test values outside the normal range of age and chromosomal sex are given in bold.

**Table 2 ijms-21-08554-t002:** Genetic findings of studied patients. Specific variants in *NR5A1* and in other disorders of sex development (DSD)-related genes assessed by a DSD-tailored gene panel are listed [15].

Case	Genetic Approach	*NR5A1* Gene Variant		Second Suspected Gene Variant by NGS		Genes Studied by Sanger (Normal)
		Genomic and Protein Change †	Familial Studies	Gene Variant † and Classification ‡	Familial Studies	
1	CGA + TGP	c.88T>A; p.Cys30Ser	Non-carriers: father, mother, brother	*STAR*, c.361C>T; p.Arg121Trp (rs34908868)/VUS ‡	Carrier: fatherNon-carriers: mother, brother	*SRD5A2*, *AR*, *FMR1*
2	CGA + TGP	c.614_615insC; p.Gln206ThrfsX20. Ass with 46,XY DSD [7]	Non-carrier: mother	*AMH*, c.428C>T; p.Thr143Ile. Ass with PCOS [19]/VUS ‡	Non-carrier: mother	*AR*, *HSD17B3*
3	TGP	c.250C>T; p.Arg84Cys. Ass with 46,XY gonadal dysgenesis [17]	Carriers: mother, uncle, aunt, grandfather, brother. Non-carriers: father, sister, grandmother	Further studies needed		No
4	CGA	c.910_913delGAGC; p.Glu304CysfsX26	Carrier: mother. Non-carrier: sister	ND		*SRY*, *AR*
5	CGA + TGP	c.902G>A; p.Cys301Tyr	Carrier: mother (mosaicism). Non-carriers: father, brother	Further studies needed		*SRY*, *HSD17B3*
6	CGA + TGP	c.71A>T; p.His24Leu	ND	Further studies needed		*AR*
7	TGP	c.1183_1185delGAG; p.Glu395del	Carrier: mother. Non-carrier: father	*ZFPM2*, c.1632G>A; p.Met544Ile. Ass with 46,XY DSD [18]/LP ‡	Carrier: fatherNon-carrier: mother	No

Ass, associated; CGA, candidate gene approach; LP, likely pathogenic; ND, not determined; PCOS, polycystic ovary syndrome; TGP, targeted gene panel; VUS, variant of unknown significance. † If mutation has been previously reported, reference is indicated; ‡ According to ACMG (American College of Medical Genetics) standards [16]. Sequence information is based on the following reference sequences: *NR5A1*: NM_004959; *STAR*: NM_000349; *AMH*: NM_00479 and *ZFPM2*: NM_012082. All variants were present in heterozygous form.

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
