# Peer review of "Variants of *STAR, AMH* and *ZFPM2/FOG2* May Contribute towards the Broad Phenotype Observed in 46,XY DSD Patients with Heterozygous Variants of *NR5A1"

_ijms, 2020, doi:10.3390/ijms21228554_

Round 1
Reviewer 1 Report
This paper by LaPiscina et al describes 7 patients with NR5A1 variants linked to a sex development disorder, developing a genotype-to-phenotype correlation map for the gene. This manuscript is an outstanding start to something that advances the literature of DSD. With a few changes this could be an outstanding paper!
Paper could use some comparisons to the other known missense variants of NR5A1 such as ClinVar or Geno2MP and maybe even gnomAD/TOPMed population level variants. Would elevate the comparison of these 7 patients into the higher context of DSD. As repeatedly claiming to develop a genotype-to-phenotype correlation, seven patients alone is not sufficient to answer this complex issue. So authors could either tone back the genotype-to-phenotype language or they could expand with some comparative work to the broader NR5A1 knowledge.
Line 101 I think authors are referring to pathogenic relative to deleterious. Deleterious refers to the protein outcome of which ACMG rules do not annotate. Pathogenic is used for the clinical ACMG description of disease causing. Table 2 should have the clinical variant annotation from ACMG rules (pathogenic, likely pathogenic, VUS). In addition, lines 122-130 also need clinical annotations of the secondary variants. As the methods state they followed ACMG guidelines the authors should provide a table with the ACMG criteria met for pathogenicity.
Are there any expression quantitative trait loci (eQTLs, GTEx) for genes such as STAR, AMH and ZFPM2/FOG2 that could modulate phenotype but are not discussed in the current manuscript?
Throughout manuscript it would be nice to normalize the use of patient vs proband vs individual as they are all used even in the same paragraph (ex 101-121).
Some small grammatical errors throughout that could use some final proof read.
“Western” blot is lowercase (line 152). Only Southern is capital as that was a name or if the others are first word of sentence.
Reviewer 2 Report
The article “Variants of STAR, AMH and ZFPM2/FOG2 may
contribute towards the broad phenotype observed in 46,XY DSD patients with heterozygous variants of NR5A1” is well written. The aim of the study is rather clear. The study has a good design. There is a large number of tables and figures of good quality presented in the manuscript. The interpretation of data obtained from literature sources is logical and reliable.
The manuscript can be recommended for publication in the journal after minor revision.
It is recommended to insert a list of abbreviations into the article.
It is recommended to add links to articles of 2019-2020 in chapter "References".
